



# Salinity as a key control on the diazotrophic community composition in the Baltic Sea

Christian Furbo Reeder[1], Ina Stoltenberg[1], Jamileh Javidpour[1], Carolin Regina Löscher[1,2]

[1]Department of Biology, University of Southern Denmark, Campusvej 55, 5230 Odense M, DK

[2]Danish Institute for Advanced Study, University of Southern Denmark, Campusvej 55, 5230 Odense M, DK

Correspondence should be addressed to crfurbo@biology.sdu.dk

**Abstract.** Over the next decade, the Baltic Sea is predicted to undergo severe changes including a decrease in salinity due to altering precipitation. This will likely impact the distribution and community composition of Baltic Sea $N_2$ fixing microbes, of which especially heterocystous cyanobacteria are adapted to low salinities and may expand to waters with currently higher

salinity, including the Danish Strait and Kattegat, while other high-salinity adapted $N_2$ fixers might decrease in abundance.

In order to explore the impact of salinity on the distribution and activity of different diazotrophic clades, we followed the natural salinity gradient from the Eastern Gotland and Bornholm Basins through the Arkona Basin to the Kiel Bight and combined $N_2$ fixation rate measurements with a molecular analysis of the diazotrophic community using the key functional marker gene for $N_2$ fixation *nifH*, as well as the key functional marker genes *anf* and *vnf,* encoding for the two alternative

nitrogenases.

We detected $N_2$ fixation rates between 0.7 and 6 nmol N $L^{-1}$ $d^{-1}$, and the diazotrophic community was dominated by the cyanobacterium *Nodularia* and the small unicellular, cosmopolitan cyanobacterium UCYN-A. *Nodularia* was present in abundances between 8.07 x $10^5$ and 1.6 x $10^7$ copies $L^{-1}$ in waters with salinities of 10 and below, while UCYN-A reached abundances of up to 4.5 x $10^7$ copies $L^{-1}$ in waters with salinity above 10. Besides those two cyanobacterial diazotrophs, we

found several clades of proteobacterial $N_2$ fixers and alternative nitrogenase genes associated with *Rhodopseudomonas palustris*, a purple non-sulfur bacterium. Based on statistical testing, salinity was identified as the primary parameter describing the diazotrophic distribution, while pH and temperature did not have a similarly significant influence on the diazotrophic distribution. While this statistical analysis will need to be explored in direct experiments, it gives an indication for a future development of diazotrophy in a freshening Baltic Sea with UCYN-A retracting to more saline North Sea waters

and heterocystous cyanobacteria expanding as salinity decreases.





## 1 Introduction

The Baltic Sea (Fig. 1) is a marginal, brackish sea characterized by a natural salinity gradient increasing from the North-East
to the South-West. The Baltic Sea covers an area of 415000 km$^2$ with a permanent halocline preventing vertical mixing,
oxygen ($O_2$)-depleted waters in the deeper basins and coastal systems, accompanied with the occasional accumulation of
hydrogen sulfide ($H_2S$) and ammonium ($NH_4^+$) below the chemocline (Conley et al., 2002; Hietanen et al., 2012; Lennartz et
al., 2014). Prior studies have shown that the Baltic Sea experienced a 10-fold increase in $O_2$-depleted areas over the last 115
years, covering an area of 12000 – 70000 km$^2$ making it one of the ocean areas most severely affected by deoxygenation
(Reusch et al., 2018). Between 1871 and 2013, the Baltic Sea showed an increase in temperature of 0.1°C decade$^{-1}$ exceeding
the average global trend of about 0.06°C decade$^{-1}$ (Reusch et al., 2018; Rutgersson et al., 2014) and decreased salinity (1-2
kg$^{-1}$) (Liblik and Lips, 2019). Nitrogen (N) deposition rates are now among the highest in marine areas (Reusch et al., 2018).
The resulting eutrophication affects severely sensitive coastal areas resulting in high pelagic production, frequent events of
anoxia, and decreased biodiversity (Breitburg et al., 2018; Carstensen et al., 2014; Maar et al., 2016; Reusch et al., 2018;
Rutgersson et al., 2014). In this context, the Baltic Sea has been described as a "time machine" for how future oceans will
respond to climate change (Reusch et al., 2018), making it an ideal environment to investigate biological fixation of
dinitrogen gas ($N_2$ fixation) in response to such changes.

$N_2$ fixation is the primary external source of new N to life in the ocean and plays a crucial role for primary production,
and thus carbon dioxide ($CO_2$) uptake from the atmosphere (Capone, Douglas G; Carpenter, 1982; Gruber, 2005).
Specialized organisms, called diazotrophs, carry out this highly energy-demanding process. The diversity of diazotrophs is
not easy to describe, as those organisms spread across the microbial tree of life, and are found in both, the bacterial and
archaeal domains (Zehr et al., 1998). $N_2$ fixation is catalyzed by an enzyme complex, the nitrogenase, which exists in three
different subtypes: the molybdenum (Mo)-iron (Fe) type referred to as Nif, the vanadium (V)-Fe type called Vnf, and a Fe-
Fe type called Anf (Chisnell et al., 1988; Robson et al., 1986). The different metal cofactors are essential in the context of
the redox sensitivity of the three nitrogenases, respectively. Mo is preferentially available in the presences of at least traces
of $O_2$, contrary to V and Fe, which are more soluble and thus available under anoxic conditions (Bennett and Canfield, 2020;
Bertine and Turekian, 1973; Crusius et al., 1996; Dixon and Kahn, 2004; Morford and Emerson, 1999). The differential
availability of those trace metals along redox gradients might have played a role in the evolutionary development of those
nitrogenase types (Anbar, 2008; Mus et al., 2019) and their distribution might change in a future ocean impacted by
acidification, deoxygenation (Keeling et al., 2010; Loescher et al., 2014; Schmidtko et al., 2017; Stramma et al., 2008) and
desalination (Olofsson et al., 2020). To date, most studies focused mainly on Nif-type nitrogenases in the marine
environment, while Vnf and Anf nitrogenases were often overlooked. Molecular detection of the *nifH* gene does not fully
recover the diversity of *vnfH* and *anfH,* possibly resulting in an underestimation of alternative nitrogenases in the
environment (Affourtit et al., 2001; Farnelid et al., 2009, 2013; Man-Aharonovich et al., 2007; Steward et al., 2004; Zehr et
al., 1995). Nonetheless, the few available studies focusing on alternative nitrogenases identified them to be abundant and





diverse and suggested to play a role for $N_2$ fixation in various ecosystems, including the upper water column of the ocean, $O_2$-depleted waters, and Baltic Sea sediments (Bellenger et al., 2011, 2014; Betancourt et al., 2008; Christiansen and Löscher, 2019; Farnelid et al., 2009; Loveless et al., 1999; McRose et al., 2017; Tan et al., 2009; Zehr et al., 2003; Zhang et al., 2016).

In the Baltic Sea, $N_2$ fixation is considered to take place in sunlit surface waters mainly and carried out by three genera of heterocystous cyanobacteria, *Aphanizomen sp, Nodularia spumigena* and *Anabaena sp.* (now referred to *Dolichosperum sp.* (Wacklin et al., 2009)), all of them containing the classic Nif-nitrogenase (Janson et al., 1994; Leppanen et al., 1988), and contributed massively to $N_2$ fixation with rates of up to $82 – 191$ nmol N $L^{-1}$ $d^{-1}$ (Larsson et al., 2001). At the same time, cyanobacteria consume only a fraction of the N they fix with estimates in the range of $2.19 – 7.74$ nmol N $L^{-1}$ $d^{-1}$ (Janson et

al., 1994; Larsson et al., 2001; Wasmund, 1997). More recent studies, based on cell-specific measurement of *Aphanizomen sp* and *Nodularia spumigena*, revealed that they release up to 35% of fixed N, translating into a substantial fraction of fixed N leaking out, available for primary production (Ploug et al., 2010, 2011)  In addition, low rates of $N_2$ fixation ($0.44$ nmol N $L^{-1}$ $d^{-1}$) have been described in anoxic waters of the Baltic Sea (Farnelid et al., 2013). These rates were accompanied by a highly diverse Nif-containing heterotrophic diazotroph community at and below the chemocline in anoxic $NH_4^+$-rich waters

(Farnelid et al., 2013). Heterotrophic diazotrophs were closely related to those in other $O_2$-depleted marine systems regions including the eastern tropical South (ETSP) and North Pacific (ETNP), the Arabian Sea (AS), and an anoxic basin in the Californian Bight (Fernandez et al., 2011; Hamersley et al., 2011; Jayakumar et al., 2012, 2017; Loescher et al., 2014). Non-cyanobacterial diazotrophs have been found to fix $N_2$ actively. However, it is still inconclusive what regulates and controls their $N_2$ fixation activity.

Over the last three decades, the Baltic Sea has experienced a trend of freshening in the range of $1-2$ $kg^{-1}$ in surface waters (Liblik and Lips, 2019) or between $0.4$ to $1.2$ (Saraiva et al., 2019). Earlier studies suggested that salinity impacts the diazotrophic community composition and the activity of nitrogenases in pure cultures of *Azotobacter* sp. marine microbial mats and estuaries (Dicker and Smith, 1981; Marino et al., 2006; Severin et al., 2012). Surface salinity in the Baltic Sea ranges between 0 and 10 (Fig. 1) making it one of the major drivers for microbial composition and distribution in the Baltic

Sea (Dupont et al., 2014; Olofsson et al., 2020; Wulff et al., 2018). Typically, *Aphanizomenon sp.* dominate the northern part (e.g. the Bothnian Sea) with a salinity of 0-2, while *Nodularia spumigena* prefer the southern part (e.g. the Southern Baltic Proper) with a salinity of 8-10 (Lehtimäki et al., 1997; Rakko and Seppälä, 2014). These observations speak for potential changes in the cyanobacterial composition in future freshening events (Olofsson et al., 2020; Wasmund et al., 2011). To now explore the impact and importance of salinity for diazotrophy in the Baltic Sea, we carried out a survey

including chemical profiling, rate measurements, and molecular genetic mining in a low productive fall season. In addition, we compared our data to available datasets to explore controls on Baltic Sea $N_2$ fixation and to be able to predict the future distribution of Baltic Sea diazotrophs and of $N_2$ fixation rates.





## 2 Material and Methods

### 2.1 Seawater sampling

Samples were collected during the cruise AL528 on the German research vessel RV Alkor to the Baltic Sea from 17.09.2019 to 28.09.2019 along a transect through the major basins of the Baltic Sea (see Fig. 1 for a cruise plan). Specifically, samples were collected at a station in the Kiel fjord (KB06), one in the Arkona basin (H21), four stations in the Bornholm Basin (BB08, BB15, BB23, BB31) and the Eastern Gotland Basin (GB84, GB90a, GB107, GB108), respectively, at water depths between 3 to 115 meters (m) using a 10 L Niskin bottle rosette equipped with a conductivity- temperature- depth (CTD) sensor. Water for dissolved inorganic nitrogen (DIN) was collected from the Niskin bottles, filtered through an acid-washed 0.8 µm cellulose acetate filter (Sartorius™), using a syringe and stored in 20 ml scintillation vials at -20°C until analysis. Samples for $PO_4^{3-}$ analysis were filtered through a 200 µm mesh to avoid contamination with large zooplankton. Samples were filled in 20 ml scintillation vials and stored at -80°C until analysis. DIN and $PO_4^{3-}$ samples were analyzed on a SKALAR SAN$^{plus}$ analyzer (Thermo Fischer, Waltham, US) according to Grasshoff (1999) with a detection limit of 0.05 µmol N L$^{-1}$ for DIN and 0.03 µmol P L$^{-1}$ for $PO_4^{3-}$. Nutrient samples were taken in triplicates, handled with nitrile gloves and all equipment was rinsed with MilliQ water in between stations to avoid contamination. Samples for dissolved inorganic carbon (DIC) were collected by filling triplicates of 12 mL exetainers bubble-free at stations KB06, H31, H21 and BB15. Samples were fixated with 20µl of a saturated mercury chloride solution and stored at room temperature until analysis. DIC measurements were carried out using a 2 mM NaHCO$_3$ solution as standard and as described previously (Hall and Aller, 1992). Water samples for DNA extraction were collected from Niskin bottles. 0.5 to 1 L of seawater were immediately filtered onto a 0.22 µm pore size membrane filter (Millipore, Bilaterica, USA; exact seawater volumes were constantly recorded), and filters were stored at -80˚ C until further analysis.

### 2.2 Molecular methods

For nucleic acid extraction, filters were flash-frozen in liquid N and subsequently crushed. DNA was purified with the MasterPure Complete DNA and RNA purification Kit (Lucigen, Wisconsin, US) according to the manufacturer's protocol, with the minor modification of using 1 mL of lysis buffer to completely cover the filter pieces. RNA was purified using Qiagen AllPrep DNA/RNA Mini Kit (Qiagen, Hilden, Germany). The remaining DNA was removed with the gDNA wipeout mix, and a cDNA library was constructed using the QuantiTect Reverse Transcription Kit (Qiagen, Hilden, Germany) with the supplied RT primer set. The nucleic acid concentration and quality were checked spectrophotometrically on a MySpec spectrofluorometer (VWR, Darmstadt, Germany).

A total of 15 DNA samples were amplified for *nifH* using nested PCR with primers and PCR conditions as described in Zani et al. (2000). *anf/vnfD* were amplified using a nested PCR with primers and conditions as described in Bellenger et al. (2014) and McRose et al. (2017). Amplicons were TOPO TA-cloned (Topo TA cloning Kit for sequencing, Thermo Fisher



Scientific, Waltham, US) and Sanger-sequenced. Sequencing was carried out at the Institute of Clinical Molecular Biology in Kiel, Germany.

Sequences were quality checked and trimmed with BioEdit (Hall, 1999). *nifH* sequences were trimmed to 321 bp and *vnf/anfD* to 591 bp. This resulted in 182 *nifH* and 69 *vnf/anfD* amplicon sequences. Sequences were BLASTX-searched against the NCBI database. A reference library was created, and sequences with lower than 80% identity were discarded. Sequences were aligned in MEGAX (Kumar et al., 2018), and a maximum likelihood tree was constructed using 1000 bootstraps. To explore the overall microbial diversity, 23 samples from Kiel Fjord (station KB06, depth 5 and 10 m)), Arkona Basin (station H2, depth 4, 12, 42 m and station H31, depth 4 and 12 m), Bornholm Basin (Station BB23, depth 4, 48, 85 m, station BB15, depth 7, 41, 65 m and station BB08, 5 and 10 m) and East Gotland Basin (Station GB84, depth 9, 29, 110 m, station GB107, depth 8, 40 and 112 m and station GB108, depth 10 and 115 m) were sent to amplicon sequencing of the 16s rDNA V1V2 region using Illumina HiSeq technology. Sequencing was carried out by the Institute of Clinical Molecular Biology in Kiel, Germany. Raw reads were trimmed, quality filtered in dada2 (v1.18) (Callahan et al., 2016), and taxonomically annotated using SILVA database (v138) (Quast et al., 2013). Visualization of the cyanobacterial distribution was accessed with *phyloseq* (1.34.0) (McMurdie and Holmes, 2013).

Quantitative real-time PCRs were carried out targeting *nifH* clade-specifically for *Nodularia*, UCYN-A and gamma-proteobacterial diazotrophs (Gamma AO, Gamma PO) as described previously (e.g., Boström et al., 2007; Langlois et al., 2008; Loescher et al., 2014). As standards, serial dilutions of plasmids ($10^7$ to $10^1$ copies) containing the target *nifH* genes were used. Samples, standards, and non-template controls were run in duplicates on a Biorad qPCR machine (Biorad, Hercules, USA), and reactions were considered uncontaminated if no amplification could be detected in non-template controls after 40 qPCR cycles. To further ensure that no DNA contamination was present in our cDNA, we ran additional qPCR reactions of RNA samples. No amplification was observed. Sequences were submitted to GenBank with accession numbers MZ063808-MZ063873 and MZ063874-MZ064057 and 16S rDNA with accession number SAMN20309768-SAMN20309783.

### 2.3 Statistical Methods

Principle component analyses (PCA) on qPCR and environmental data were performed in R using the *vegan* package (Oksanen et al., 2020; R Foundation for Statistical Computing, 2017). The *simper* function (from *vegan* package) was used to identify parameters with the highest impact on diazotroph abundance. A principal component analysis (PCA) was subsequently performed using *prcomp* and visualized with the *factoextra* package (Kassambara and Mundt, 2020).

### 2.4 $^{15}N_2$ and $^{13}C$ seawater incubations

$N_2$ and C fixation rates were determined using stable isotope labelling at KB06, H21, BB15, BB08 and GB107, at water depths between 3 and 41 m. We used the bubble addition method (Montoya et al., 1996) to ensure comparability to previous studies from the Baltic Sea. Triplicate water samples were filled from Niskin bottles into 2.4 L glass bottles (Schott-Duran,





Wertheim, Germany). To each incubation bottle, 1 mL $^{15}N_2$ gas ($^{15}N_2$ 98%+ Lot#I-16727, Cambridge Isotope Laboratories, Inc., USA) and 1 mL $H^{13}CO_3$ (1g 50 mL$^{-1}$, Sigma-Aldrich, Saint Louis Missouri US) was added through an air-tight septum

and bottles were inverted to ensure mixing. The final concentration was 0.05% $^{15}N_2$ and 10 µg mL$^{-1}$ $H^{13}CO_3$. After an incubation time of 24 h, samples were filtered onto pre-combusted GF/F filters (GE Healthcare Life Sciences, Whatman, USA). A parallel set of triplicate samples was collected from each sampling depth for determining the natural abundance of N and C isotopes in the particulate organic material. Filters were stored at -20ºC until further analysis. Filters were then acidified, dried, and analyzed on an Elemental Analyzer Flash EA 1112 series (Thermo Fisher), coupled to an isotope ratio

mass spectrometer (Finnigan Delta Plus XP, Thermo Fisher).

## 3 Results and Discussion

### 3.1 Hydrochemistry

During the cruise, surface waters above 40 m water depth exhibited the typical salinity gradient with salinities decreasing from South-West to North-East (10 – 24 in the Kiel and Arkona Basin and 7-9 in the Bornholm and Eastern Gotland Basins),

accompanied by lower temperatures of 13-14ºC in the Bornholm and Eastern Gotland Basin compared to slightly higher temperatures of 15-16ºC in the Kiel Fjord and Arkona Basin (Fig. 2). A thermocline was observed at 45 m in the Bornholm Basin and 65 m in the Eastern Gotland Basin. The coastal station we sampled in the Arkona basin showed a similar stratification pattern (Fig. 2) with a salinity of 10 on the surface and 17 below 30 m water depth. This stratification pattern is typical for the region and time of the year and could also be observed from annual data from the International Council for the

Exploration of the Sea (ICES Copenhagen, 2020) (Fig. 1, B). Surface waters were well-oxygenated with $O_2$ concentrations above 156.31 µmol L$^{-1}$ in the top 40 m of the water column. Low $O_2$ water masses with concentrations below 15 µmol L$^{-1}$ were only observed in the Bornholm basin in waters deeper than 65 m (Fig. 2).

DIC concentrations were lower in the upper water column of the Eastern Gotland and Bornholm Basin with concentrations of around 1.6 mmol L$^{-1}$ and increased towards the south-western part of the cruise track. An opposite trend was visible in nutrient distribution with $NO_x$ (nitrite + nitrate) and phosphate ($PO_4^{3-}$) decreasing from the North-Eastern part

of the cruise track towards the South-Western part with surface water concentrations of 9.04 µmol L$^{-1}$ for $NO_x$ and 2.98 µmol L$^{-1}$ for $PO_4^{3-}$ in surface waters of the Eastern Gotland Basin and 0.27 µmol L$^{-1}$ for $NO_x$ and 0.18 µmol L$^{-1}$ for $PO_4^{3-}$ in the Kiel Fjord (Fig. 2). The observed nutrient concentrations were in the range of the Helsinki commission (HELCOM) dataset containing data from 2015 to 2020 and the HELCOM core indicator report (Helcom, 2018). The detected somewhat

higher nutrient concentrations in the Bornholm and Eastern Gotland Basins could result from a decaying phytoplankton bloom, decreased microbial activity or increased eutrophication. High chlorophyll concentrations in the Bornholm basin (3 µM Chl a in August and 2 µM Chl a in September), derived from the HELCOM dataset (from same areas as this cruise), indeed give evidence of a decaying phytoplankton bloom in the Bornholm Basin in September 2019 (Fig. 3). These values





are in range with the HELCOM indicator report with an average of 4 µM Chl a from June to September, and a previous
study presenting a range between 2 and 6 µM Chl a (Helcom, 2018; Suikkanen et al., 2010).

Our dataset indicates an N:P ratio well below the Redfield ratio of 16:1 in surface waters along the cruise track, with
excess P classically supposed to promote $N_2$ fixation. However, DIN is still available to the phytoplankton and microbial
community as supported by a positive intercept of the trendline, which might be considered unfavorable for $N_2$ fixation (Fig.
4). Remaining $NO_x$ concentrations in the surface waters above 40 m thus speak for a limitation of primary production by
either micronutrients, temperature, or light availability and less for a limitation by N or $PO_4^{3-}$ (Arrigo, 2005; Saito et al.,
2008). Nitrogen repletion is supported by POC: PON ratios of 3.6 to 4.94 in the Kiel Fjord and Arkona Basin and 2.92 to
6.15 in the Bornholm and Eastern Gotland Basins (Fig. 4), which are close to the Redfield ratio (Redfield, 1934). Altogether,
nutrient concentrations do not strongly support a niche for $N_2$ fixation.

**3.2 Nitrogen fixation**

While no clear niche for $N_2$ fixation could be identified, we still detected $N_2$ fixation rates in our samples from water depths
between 3 and 41 m (Fig. 5). $N_2$ fixation rates were between 0 to $6 \pm 4.73$ nmol N $L^{-1}$ $d^{-1}$ in the Arkona Basin, $0.7 \pm 0.97$ to
$27.3 \pm 38.21$ nmol N $L^{-1}$ $d^{-1}$ in Bornholm Basin and $0.67 \pm 0.51$ to $2 \pm 0.4$ nmol N $L^{-1}$ $d^{-1}$ in the Eastern Gotland Basin (Fig.
5). In Kiel Fjord, $N_2$ fixation rates were below the detection limit of our method. One outlier was observed in the Bornholm
Basin at a water depth of 5 m, with an $N_2$ fixation rate of 27.3 nmol N $L^{-1}$ $d^{-1}$. This high rate was only detected in one out of
our three replicates and was not included in our further analysis. Without this outlier, a conservative estimate of $N_2$ fixation
in the Bornholm Basin would be between $0.7 \pm 0.97$ and $5.27 \pm 1.37$ nmol N $L^{-1}$ $d^{-1}$. Though these rates are at the lower end
of previous observations from the Baltic Sea (Table 1), they are comparable to a previous study from the same region, but
not the same season, with $N_2$ fixation rates of $7.6 \pm 1.76$ nmol N $L^{-1}$ $d^{-1}$ from July and August (Farnelid et al., 2013). N2
fixation activity is supported by the $\delta^{15}$N -PON in our control samples which was in the range of control samples was found
to be 0.3 to 2 ‰ in samples from surface waters above 40 m from the Eastern Gotland and Bornholm Basins (Fig. 5). Values
between -2 to +2 ‰ are considered indicative for $N_2$ fixation (Dähnke and Thamdrup, 2013; Delwiche and Steyn, 1970).

Seasonal variations might explain the observed lower $N_2$ fixation rates if compared to previous studies. $N_2$ fixation rates
are generally described sensitive to low temperatures (Brauer et al., 2013; Church et al., 2009; Englund and Meyerson, 1974;
Moisander et al., 2010; Stal, 2009) with the energetic costs of $N_2$ fixation sharply rising at temperatures below 21 ℃ (Brauer
et al., 2013). During our cruise, surface temperatures were between 12 ℃ and 16 ℃ and would most likely play a role in the
low rates observed. Yet, growth of *Nodularia spumigena* has been demonstrated at 4 ℃ suggesting a relevance during
wintertime in the Baltic Sea (Olofsson et al., 2019). Additionally, lower light intensities could directly influence the
abundance of cyanobacterial diazotrophs in the Baltic Sea (Dera and Woźniak, 2010; Staal et al., 2002).

$N_2$ fixation rates were accompanied by low C fixation rates (Fig. 5), which were, however, increasing over depth, with an
average C fixation of $29.16 \pm 10.89$ nmol C $L^{-1}$ $d^{-1}$ in Kiel Fjord, $42.62 \pm 7.86$ nmol C $L^{-1}$ $d^{-1}$ the Arkona Basin, $40.52 \pm$





13.16 nmol C L$^{-1}$ d$^{-1}$ in Bornholm Basin and 58.52 ± 15.42 nmol C L$^{-1}$ d$^{-1}$ in Eastern Gotland Basin. Assuming Redfield stoichiometry, N$_2$ fixation sustained between 8.8 and 108% with an average of 43% of C fixation during our cruise. These rates are comparable with the contribution of N$_2$ fixation to C fixation in other oceanic regions, including the Atlantic (20-25%) and Pacific Ocean (1-4%), the Mediterranean Sea (8%), and the Bay of Bengal (<1%) (Church et al., 2009; Dore et al.,
2002; Fonseca-Batista et al., 2019; Rahav et al., 2013; Saxena et al., 2020; Tang et al., 2019) (Table 2). The high contribution to primary production indicates that N$_2$ fixation may promote or extend primary production into the fall season if only at low rates.

### 3.3 Diazotrophic community composition

We assessed the diazotrophic diversity based on *nifH* and *vnf/anfD* amplicons (Fig. 6 and 7). Sequences belonging to
*Nodularia spumigena* (19%) and UCYN-A (26%) were dominant in the sequence pool. While   *Nodularia* clades are classically abundant in the Baltic Sea waters (Boström et al., 2007; Farnelid et al., 2013; Klawonn et al., 2016; Larsson et al., 2001), only one previous report of UCYN-A from the Baltic Sea exists, where it had been detected in waters of the Danish Strait and the Great Belt (Bentzon-Tilia et al., 2015). Moreover, we detected *Pseudanabaena*-like sequences representing 8 % of the sequence pool consistent with previous studies (e.g. Acinas et al., 2009; Farnelid et al., 2013; Klawonn et al., 2016;
Stal et al., 2003). Only 1.5 % of the sequences belonged to *Aphanizomenon sp.,* a diazotroph known to be adapted to waters with extremely low salinity (0-2) as present in the Northern part of the Baltic Sea (Lehtimäki et al., 1997; Rakko and Seppälä, 2014) and below the salinity observed during our cruise (>8).

Notably, based on our *nifH* sequence analysis, we did not identify, *Anabaena lemmermannii*, one of the common N$_2$ fixing cyanobacteria for the Baltic Sea (Bentzon-Tilia et al., 2015; Boström et al., 2007; Farnelid et al., 2013). However,
sequences related to *A. lemmermannii* were recovered from our 16S rDNA amplicon sequencing (Fig. S1 in supplementary). This discrepancy might speak for a bias against their specific *nifH* genes by our approach, or the clades we found in the 16S rDNA dataset might lack a *nifH* gene.

We detected *Nodularia*-like sequences (primer/probe constructed against *Nodularia spumigena* and environmental *Nodularia* clusters, see Boström et al., 2007), in abundances of 3.37 x 10$^5$ to 4.06 x 10$^7$ copies L$^{-1}$ at 3-5 m water depth and
8.64 x 10$^4$ – 4.46 x 10$^6$ copies L$^{-1}$ at water depths of 20-41 m. *Nodularia*-specific *nifH* transcript abundance decreased over depth from 4.20 x 10$^5$ (5 m) to 3.5 x 10$^2$ (20 m) transcripts L$^{-1}$ (Fig. 8). *Nodularia* followed the temperature-salinity gradient with the highest abundance (up to 1.6 x 10$^7$ copies L$^{-1}$) in waters with salinity at 10 and below (the Bornholm and Eastern Gotland basin) and lowest abundances (up 1.46 x 10$^6$ copies L$^{-1}$) in waters with a salinity above 10 (Kiel Fjord and the Arkona basin). *Nodularia*-specific *nifH* gene and transcript abundances were in the same range as previously described (10$^6$
gene copies L$^{-1}$ and 10$^5$ transcripts L$^{-1}$ (Farnelid et al., 2013)). UCYN-A was identified based on both *nifH* and 16S rDNA sequences and quantified using cluster-specific qPCR. UCYN-A was present in abundances of 3.73 x 10$^3$ – 9.97 x 10$^7$ copies L$^{-1}$, similar to the one available previous study where the abundance of *nifH* specific for UCYN-A peaked in





September ( approximately $10^6$ copies L$^{-1}$) (Bentzon-Tilia et al., 2015) (Fig. 8). The same study showed *nifH* expression (roughly $10^3$ transcript L$^{-1}$) of UCYN-A in the Baltic Sea surface waters (1 m). These transcript abundances are comparable

with those obtained from our study with transcript abundances between 2.98 x $10^2$ and 3.73 x $10^2$ transcripts L$^{-1}$ at 5 m water depth and up to 3.71 x $10^5$ transcripts L$^{-1}$ at a water depth of 20 m (Fig. 8). Notably, the above-mentioned study took place in the Danish Strait, and thus in more saline waters (salinity of 13-17).

Contrary to *Nodularia,* UCYN-A reached highest abundances (up to 4.52 x $10^7$ copies L$^{-1}$) in higher saline waters impacted by the North Sea (the Kiel Fjord and Arkona Basin) and the lowest abundance (up to 6.29 x $10^5$ copies L$^{-1}$) in low

saline waters (Fig. 8, Supplementary Figure S1). UCYN-A is increasingly found throughout the oceans, often playing an important role in N$_2$ fixation (e.g. Martínez-Pérez, C., Mohr, W., Löscher, 2016; Tang and Cassar, 2019). Over the recent years, UCYN-A has been shown to have a crucial part in N$_2$ fixation and has been identified all over the globe, from polar to tropical regions and tolerating between 12ºC and 30ºC (Gradoville et al., 2020; Harding et al., 2018; Martínez-Pérez, C., Mohr, W., Löscher, 2016; Mills et al., 2020; Short and Zehr, 2007; Tang and Cassar, 2019). Despite UCYN-A exposing

highest abundances in Baltic Sea waters with higher salinity, we found UCYN-A present and active across the salinity gradient.

Besides those cyanobacterial diazotrophs, we detected several clades of proteobacteria related to *Rhodopseudomonas palustris* (4%)*, Pelobacter sp* (9%) and gamma proteobacteria of the Gamma-AO clade (3%, Langlois et al., 2008). We quantified gamma-proteobacterial *nifH* in abundances between 1.78 x $10^4$ -1.82 x $10^5$ copies L$^{-1}$ (Gamma AO) and their

transcripts in abundances of 2.89 x $10^3$ - 3.91 x $10^3$ transcripts L$^{-1}$. Despite lack of clones clustering with Gamma PO, the group could be detected in abundances between 8.21 x $10^2$ -1.37 x $10^5$ copies L$^{-1}$ and their transcripts in abundances between 3.04 x $10^3$ - 8.05 x $10^3$ transcripts L$^{-1}$, indicating a potential contribution to N$_2$ fixation of those clades (Fig. 8). Diazotrophic proteobacteria are common in the ocean, however, their role for N$_2$ fixation is not fully understood (Benavides et al., 2018; Chen et al., 2019; Turk-Kubo et al., 2014). Diazotrophs related to Cluster III were dominated by *Desulfovibrio*-like (1.5%),

*Opitutaceae*-like (6%) and *Clostridium*-like sequences (1.5%). However, both *nifH* gene and transcript abundances were below the detection limit of our qPCR. Altogether, the gene and transcript abundances of cyanobacterial diazotrophs were three to four orders of magnitude higher than those of non-cyanobacterial clusters suggesting that heterotrophic microbes may have only played a minor role for N$_2$ fixation in Baltic Sea surface waters during our cruise.

In the pool of alternative nitrogenase sequences of the *anfD* type, we identified the purple non-sulfur bacterium

*Rhodopseudomonas palustris*. We could not recover any *vnfD* sequences, which might be an adaptation to low vanadium concentrations in the Baltic Sea (Bauer et al., 2017). Purple non-sulfur bacteria (e.g *Rhodopseudomonas palustris*) typically require anoxic conditions to fix N$_2$ (Masepohl and Hallenbeck, 2010), at the sample locations monitored during our cruise. The role of alternative nitrogenases in the environment is poorly understood, partly due to a lack of data and detection systems. Our data shows however the presence of alternative nitrogenases in the Baltic Sea surface water, which may sustain

N$_2$ fixation under certain conditions, for example in anaerobic micro-niches (Bertagnolli and Stewart, 2018; Farnelid et al.,





2019; Paerl and Prufert, 1987; Pelve et al., 2017), including sinking particles (Chakraborty et al., 2021; Pedersen et al., 2018).

### 3.4 Factors impacting the diazotrophic distribution

To identify parameters impacting on the distribution of diazotrophs during our cruise, we carried out a *simper* analysis.
Interestingly, pH and temperature were less important for describing the diazotrophic distribution. The PCA suggested that depth and salinity explained the distribution of both the diazotrophic community and $N_2$ fixation best (Fig. 9). Moreover, it revealed a correlation of $N_2$ fixation and the *Nodularia* abundance and less of a correlation between $N_2$ fixation and UCYN-A or other diazotroph clusters (Fig. 9). A decreasing salinity would thus not only promote the distribution of *Nodularia*-like diazotrophs, and decrease UCYN-A abundances, but also promote $N_2$ fixation rates in such a future scenario. Intuitively, it is
of little surprise that our data (Fig. 9) and previous studies indicate salinity to be a key control for cyanobacterial distribution and $N_2$ fixation in the Baltic Sea (Dupont et al., 2014; Olofsson et al., 2020; Rakko and Seppälä, 2014). However, a previous study based on a compiled dataset (1979-2017) and modelling approaches indicated that salinity does not affect the biovolumes of the filamentous *Nodularia spumigena* but rather species-interactions (Karlberg and Wulff, 2013). Our study cannot evaluate such interactions and their impact of $N_2$ fixation; however, our data do point towards *Nodularia* expanding
into low salinity waters in the future thus complementing UCYN-A and possibly increasing $N_2$ fixation rates in those waters.

Our analysis did not show temperature and pH to be major descriptors of $N_2$ fixation during this cruise, consistent with previous findings with elevated carbon dioxide concentrations not causing any response in $N_2$ fixation in the Baltic Sea (Olofsson et al., 2019; Paul et al., 2016; Wulff et al., 2018). Moreover, a very recent study showed that ocean acidification has an impact on the diazotroph community composition and can decrease $N_2$ fixation rates in the subtropical Atlantic Ocean
(Singh et al., 2021). Most likely, the contradicting results are due to other factors obscuring the stimulation by $N_2$ fixation (Karlberg and Wulff, 2013; Wannicke et al., 2018). Besides pH, temperature has also been suggested as a control on diazotrophy, either directly or indirectly, by impacting on $O_2$ solubility (Stal, 2009).

Besides, increasing nutrient loads in the Baltic Sea might further stimulate primary production leading to increased anoxia and $PO_4^{3-}$ release from sediments (Ingall and Jahnke, 1997), which will, in turn, fuel $N_2$ fixation and primary
production (Canfield, 2006; Saraiva et al., 2019). Filamentous cyanobacteria benefit from excess $PO_4^{3-}$ (Olofsson et al., 2016), thus, excess $PO_4^{3-}$ resulting from progressive deoxygenation might stimulate blooms of *Nodularia* (Degerholm et al., 2006; Schoffelen et al., 2018).

In case of a future freshening of the upper water column (Liblik and Lips, 2019), our data points towards $N_2$ fixation by cyanobacteria such as *Nodularia* will increase. This could lead to a scenario with increased bloom formations and expansion
of such large heterocyst cyanobacteria across the Baltic Sea, as previously observed (Kahru et al., 1994; Kahru and Elmgren, 2014). An increased organic matter load would possibly lead to enhanced respiration by heterotrophic microbes promoting the further expansion of $O_2$ depleted waters. This could lead to increased denitrification resulting in N-loss and emission of $N_2O$, fueling global warming, an increase in euxinic events as already observed in coastal Baltic waters (Breitburg et al.,


2018; Carstensen et al., 2014; Lennartz et al., 2014). Further, blooms of *Nodularia* could lead to an increased load of Baltic
Sea waters with the toxin Nodularin (Sivonen et al., 1989), which can harm Baltic Sea biota and poison animals (Main et al., 1977; Nehring, 1993) and ultimately humans by impacting e.g. fishery industry (Karjalainen et al., 2007).

## 4 Conclusion

During our cruise, we explored the diazotrophic community composition and $N_2$ fixation rates along the natural salinity gradient in the Baltic Sea in the fall season. $N_2$ fixation rates were detectable but low for the Baltic Sea and sustained by a
diazotrophic community dominated by the heterocystous cyanobacterium *Nodularia* and the small unicellular cyanobacterium UCYN-A. The two different types of cyanobacteria occupied two different niches defined by salinity ranges, respectively, with *Nodularia* dominating in lower saline waters (8-9) and UCYN-A in high-salinity waters (>10). Statistical analysis revealed that $N_2$ fixation is quantitatively mainly driven by *Nodularia* clades and that both, *Nodularia* abundances and $N_2$ fixation rates, are best explained by salinity. In the context of a predicted freshening of the Baltic Sea, the
habitat of *Nodularia*-like heterocystous cyanobacteria would extend towards the South-Western part of the Baltic Sea, possibly replacing the community of UCYN-A and increasing $N_2$ fixation in those waters. Enhanced $N_2$ fixation might facilitate primary productivity, and organic matter export to waters below the euphotic zone and could thus have severe impacts on the Baltic Sea biogeochemistry including increased respiration, $O_2$ and N loss.

## Author contribution

C. Reeder and C. Löscher designed the experiments and C. Reeder carried out the experimental work. J. Javidpour and I. Stoltenberg designed the sampling strategy during the expedition and contributed essential datasets. C. Reeder and C. Löscher prepared the manuscript with contributions from all the co-authors.

## Competing interest

The authors declare that they have no conflict of interest.

## Acknowledgement
We thank the captain and crew and the chief scientist J. Süling of Alkor 528 for their support during the cruise. We thank E. Laursen and R. Orloff Holm for their technical assistance. Funding for this study was received from the Villum Foundation (Grants #16518 to D. Canfield and #29411 to C. Löscher) and the European Union's Horizon 2020 research and innovation program (Grant agreement no. 774499 to J. Javidpour) for funding the GoJelly project. Further support was received from



the Nordcee labs at SDU and the GEOMAR- Helmholtz Centre for Ocean Research Kiel by providing access to their research vessel and facilities.

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





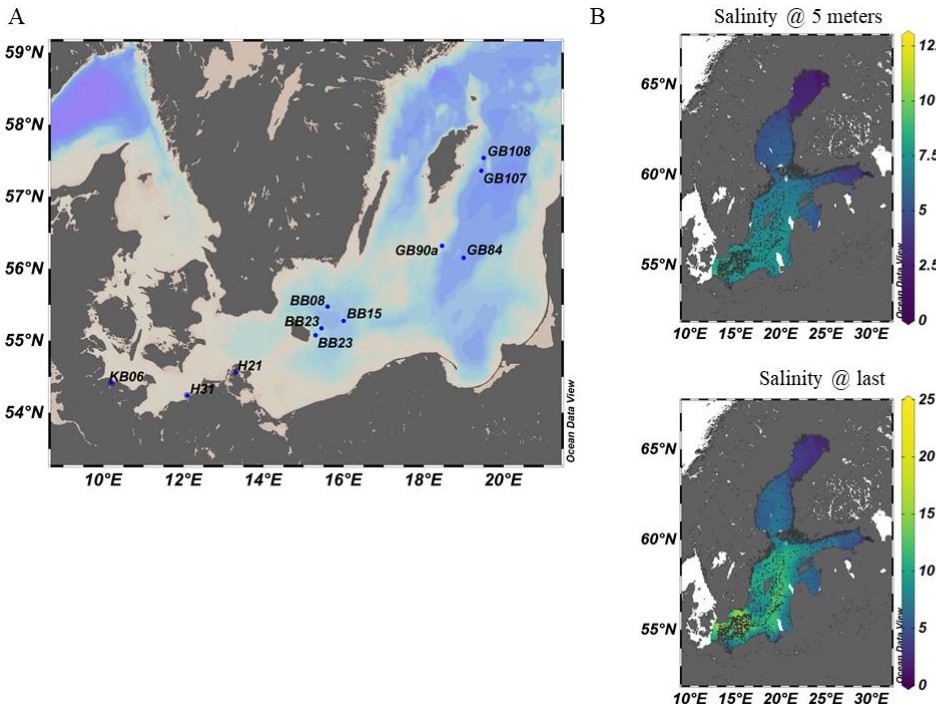

**Figure 1. Overview of cruise track and stations from this study (A) Distribution of O₂, salinity (sal) in surface water (5 m) and the deepest sample at the station (last) in the Baltic Sea. Data for figure B were obtained through ICES oceanography spanning from 2019-08-01 to 2019-10-30 (ICES Copenhagen, 2020).**





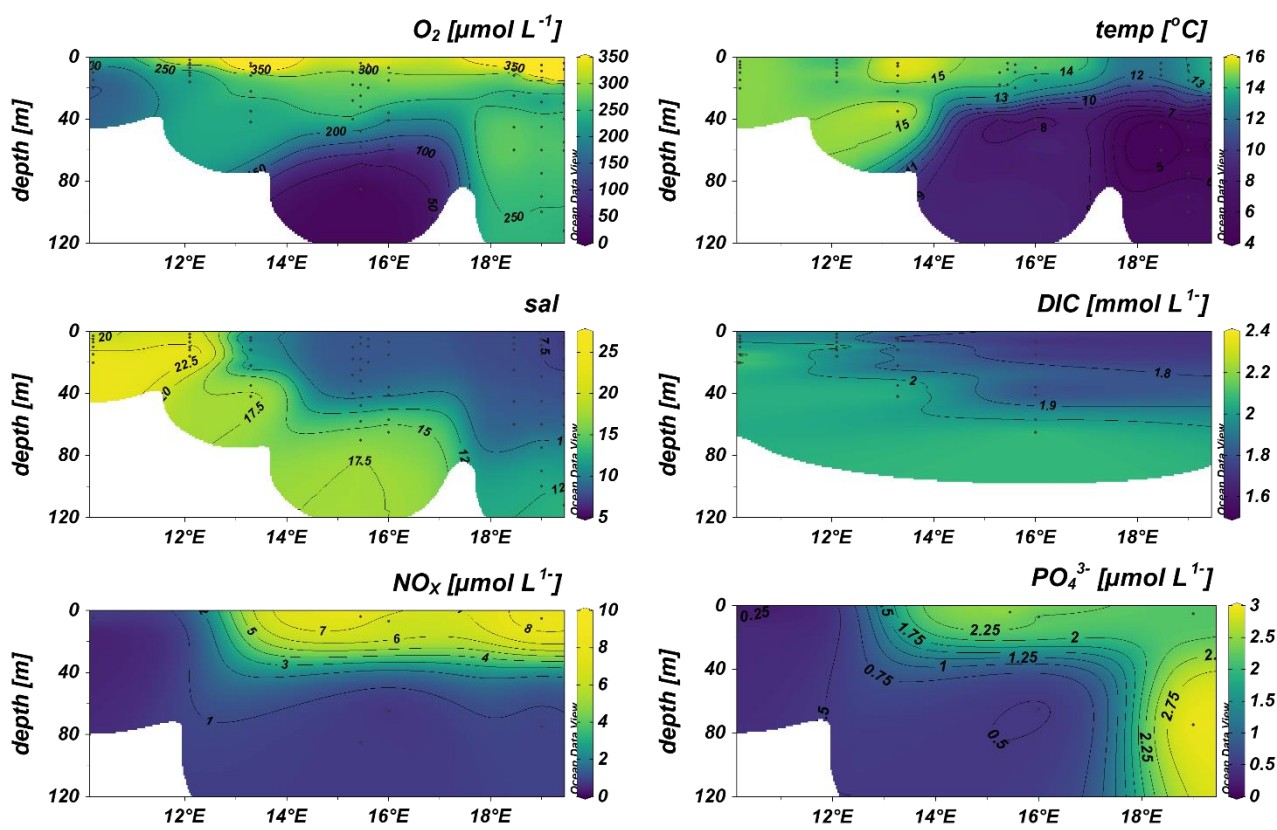

**Figure 2 Distribution of surface water O₂ (µmol L⁻¹), temperature (temp) (°C), salinity (sal), dissolved inorganic carbon (DIC) (mmol L⁻¹) and NOx (nitrate + nitrite) and PO₄³⁻ in µmol L⁻¹.**


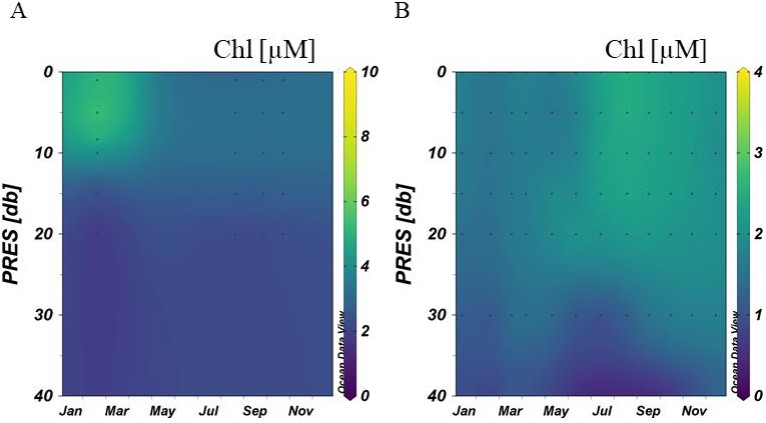

**Figure 3 Overview of chlorophyll data derived from HELCOM in (A) the Arkona Basin and (B) the Bornholm Basin. Note that data points are missing for June and July in Arkona Basin.**




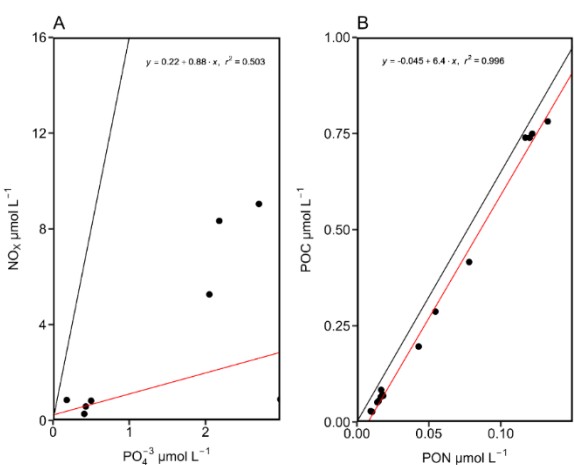

**Figure 4** N:P (A) and C:N (B) ratios from station KB06, BB23, BB15, BB08 and GB107. The black lines depict the N:P (16:1) and C:N (106:16) ratios after Redfield. The red lines indicate the trendlines of our data. An N:P ratio below the canonical Redfield ratio of 16:1 suggests a niche for N$_2$ fixation However, the positive intercept speaks for DIN still being available, which might be considered as unfavorable for N$_2$ fixation (A). A C:N ratio below Redfield might suggest a slight deficiency in N (B), this might result from degrading organic matter in the POM pool.

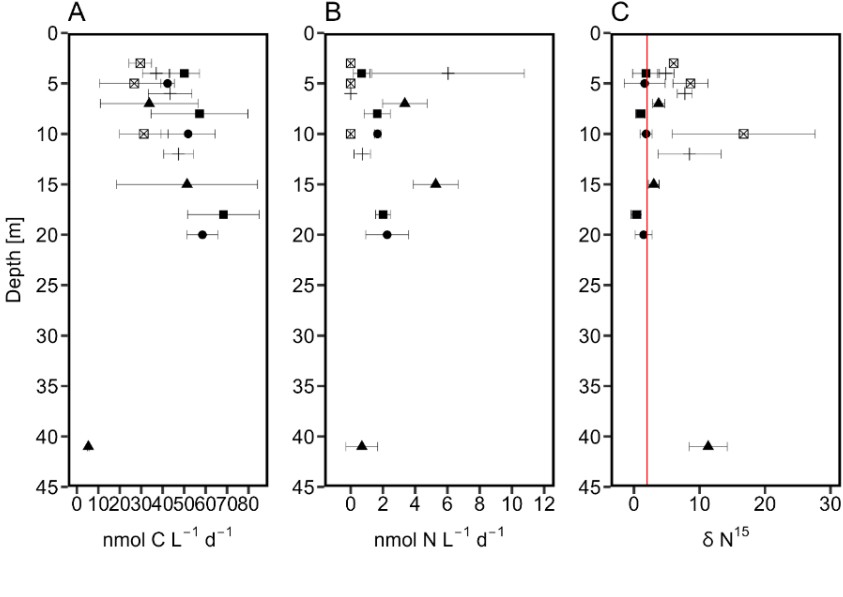

**Figure 5** C (A) and N$_2$ (B) fixation rates [nmol L$^{-1}$ d$^{-1}$]. Similar in all stations, C fixation increased over depth, followed by a decrease at a water depth of 41 m. Higher rates were observed in the Bornholm and Eastern Gotland Basins compared to the Kiel and Arkona Basins. N$_2$ fixation rates increased slightly over depth, followed by a decrease at 41 m water depth in the Bornholm Basin. (C) δ$^{15}$N [‰] from station KB06, H21, BB08, BB15 and GB107. The red line indicates a δ$^{15}$N of 2 ‰. Values below 2‰ speak for previous or current N$_2$ fixation (Dähnke and Thamdrup, 2013).





**Figure 6 Maximum likelihood tree of *nifH* amplicons (321 bp) identified by Sanger sequencing. Clusters identified are denoted as grey triangles while identified individual sequences are denoted as 'Seq + NCBI submission ID'. Cluster I cyanobacteria are shown in green, Cluster I proteobacteria are shown in blue, and Cluster III diazotrophs are shown in orange.**



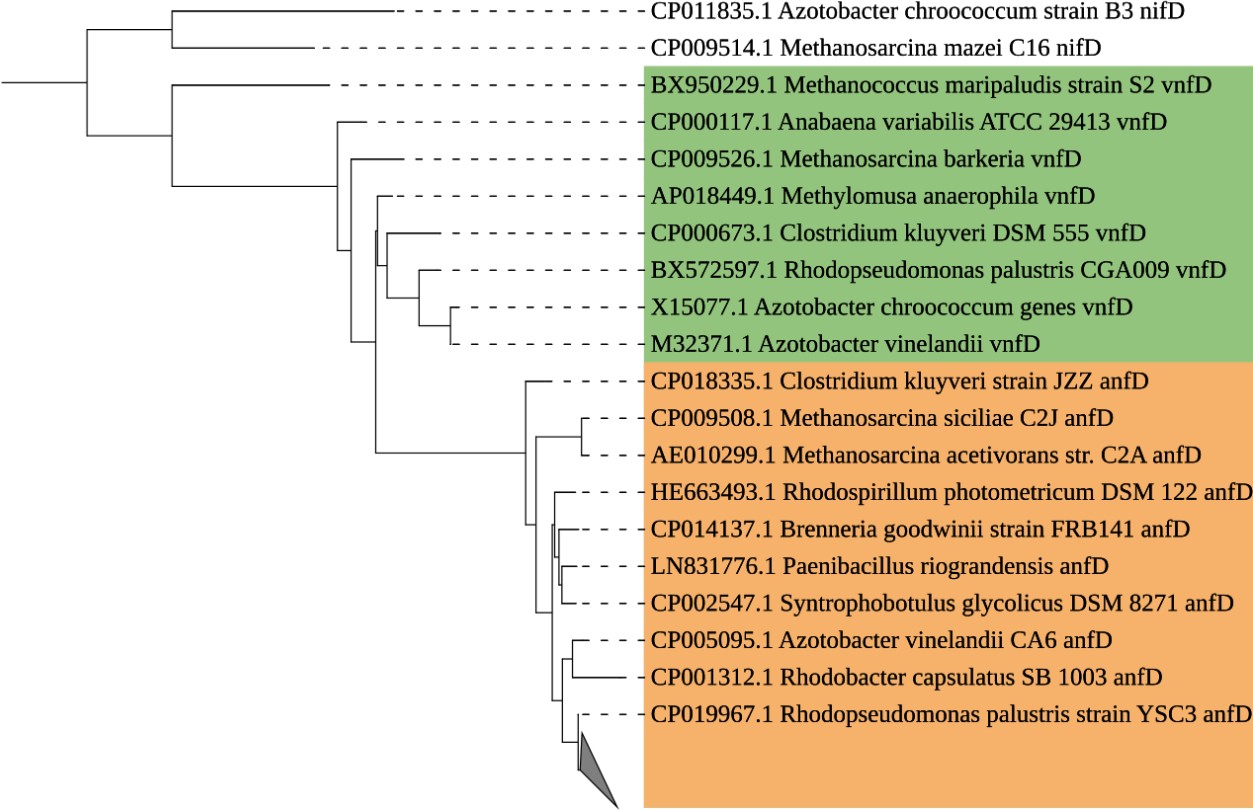

**Figure 7 Maximum likelihood tree of *vnf/anfD* amplicons (591 bp) identified by Sanger sequencing. All recovered sequences**
**clustered with *Rhodopseudomonas palustris* (grey triangle). *anfD* sequences are represented in brown, *vnfD* sequences are**
**represented in green. A *nifD* Sequence has been chosen as the outgroup.**





**Figure 8 Cluster-specific nifH (A) gene and (B) transcript abundances of Nodularia, UCYN-A and Gamma-PO/AO. In general, both cyanobacterial groups are decreasing in nifH gene abundance over depth at all stations. Transcript abundances of Nodularia-nifH were higher in surface waters (5 m), contrary to UCYN-A, which showed an opposite trend. Gamma-PO/AO gene and transcripts abundance generally increased over depth.**





Figure 9 Principal component analysis (PCA) of components one and two (A), one and three (B). Positive correlations were identified between salinity and UCYN-A, while *Nodularia* and salinity are negatively correlated. Moreover, $N_2$ fixation correlated positively with *Nodularia*. Stations are color-coded. Each circle denoted one sampling point, bigger circles indicate the center for each station.




**Table 1 Overview of N$_2$ fixation rates measured from 1972 to 2019. Calculated N$_2$ fixation rates for this study are average over 20 m water depth.**

| Date | Basin | Method | N$_2$ fixation (mmol N m$^{-2}$ d$^{-1}$) | Source |
|---|---|---|---|---|
| 1972-1974 | archipelago near Stockholm | Acetylene reduction | 0.4-1.8 | (Brattberg, 1997) |
| Jul/Aug-1976 | Askö (Sweden) | Acetylene reduction | 0.057 - 0.012 | (Lindahl et al., 1978) |
| 1977 | Öregrundsgrepen SW Bothnian Sea | Acetylene reduction | 0.01369863 | (Rinne et al., 1978) |
| | Öregrundsgrepen SW Bothnian Sea | Acetylene reduction | 0.011780822 | (Rinne et al., 1981) |
| | Mecklenburg bay | Acetylene reduction | 0.057 | |
| | Arkona Basin | Acetylene reduction | 0.286 | |
| 1978 | Bornholm Basin | Acetylene reduction | 0.286 | |
| | Gulf of Gdansk | Acetylene reduction | 0.857 | (Hübel H, 1984) |
| | E. Gotland Basin | Acetylene reduction | 0.357 | |
| 1974-1983 | Arkona Basin | Acetylene reduction | 0.047 - 0.301 | |
| | North Baltic Proper | Acetylene reduction | 0.14 - 6.3 | |
| 1980 + 82 + 84 | Center Baltic Proper | Acetylene reduction | 27.1 - 55.7 | (Niemistö et al., 1989) |
| | South Baltic proper | Acetylene reduction | 34.4 | |
| Aug-90 | E. Gotland Basin | | 2.44 | (Haupt, 1991) |
| Jul/Aug-1993 | Baltic Proper | | 0.78 - 1.42 | (Stal and Walsby, 1998) |
| 1990-1997 | Baltic proper | Calculated from | 0.21 - 2.6 | (Rahm et al., 2000) |





| | | nutrient budget | | |
|---|---|---|---|---|
| 1994-1998 | Baltic proper without Arkona | Calculated from nutrient budget | 2.3 - 5.9 | (Larsson et al., 2001) |
| Aug-97 | Baltic Proper | $N^{15}$ tracer method | 7.113333333 | |
| Oct-97 | Baltic Proper | $N^{15}$ tracer method | 0.986666667 | |
| Nov-97 | Baltic Proper | $N^{15}$ tracer method | 0.1 | |
| Feb-98 | Baltic Proper | $N^{15}$ tracer method | 0.02 | |
| Mar-98 | Baltic Proper | $N^{15}$ tracer method | 0.05 | (Wasmund et al., 2001) |
| May-98 | Baltic Proper | $N^{15}$ tracer method | 0.055 | |
| Jul-98 | Baltic Proper | $N^{15}$ tracer method | 1.23 | |
| Aug-98 | Baltic Proper | $N^{15}$ tracer method | 1.9825 | |
| Nov-98 | Baltic Proper | $N^{15}$ tracer method | 0.19 | |
| 2001 | Baltic Proper | $N^{15}$ tracer method | 0.156 | (Wasmund et al., 2005) |
| 2002 | Baltic Proper | $N^{15}$ tracer method | 0.252054795 | |
| Jul-11 | Gotland Basin | $N^{15}$ tracer method | 0.038 | (Farnelid et al., 2013) |
| 2012 | Danish Strait - Great Belt | $N^{15}$ tracer method | 0.167 | (Bentzon-Tilia et al., 2014) |
| June-Aug12 | Baltic proper | $N^{15}$ tracer method | 3.6 | (Klawonn et al., 2016) |
| June-Aug13 | Baltic proper | $N^{15}$ tracer method | 0.4 | |
| 1999-2017 | Baltic proper | biovolumes and empirical rates | 0.36 ± 0.07 | (Olofsson et al., 2021) |
| 2013-2017 | Baltic proper | biovolumes and empirical rates | 0.37 - 0.07 | |




| 1999-2004 | Bothnian sea | biovolumes and empirical rates | 0.083 | |
| 2012-2017 | Bothnian sea | biovolumes and empirical rates | 0.075 | |
| 1999-2017 | Bothnian sea | biovolumes and empirical rates | 0.059 ± 0.023 | |
| 2019 | Bornholm | N$^{15}$ tracer method | 0.15 | This study |
| | East Gotland | | 0.026 | |


**Table 2 The contribution of N$_2$ fixation to primary production (PP) in different ocean basins.**

| Location | PP Contribution (%) | Source |
| --- | --- | --- |
| North east Atlantic | 25 | (Fonseca-Batista et al., 2019; Tang et al., 2019) |
| West Northeast Atlantic | 20 | |
| Mediterranean Sea | 8 | (Church et al., 2009; Dore et al., 2002; Fonseca-Batista et al., 2019; Rahav et al., 2013; Saxena et al., 2020) |
| North Pacific Ocean | 1-4 | |
| Bay of Bengal | <1 | |
| Baltic Sea | 8-25 | This study |
