# Peer review of "Salinity as a key control on the diazotrophic community composition in the Southern Baltic Sea"

_Ocean Science, 2021_

## Author Comment (AC1)

The study contains an interesting data set with UCYN-A including both genes and potential N2 fixing transcripts in the southern Baltic Proper. However, the study was conducted a bit off season as compared to expected and the manuscript lacks an argumentation to why this study is needed? Was it late in the season on purpose and if so why?

Authors' response (AR): We thank the reviewer for considering our manuscript interesting, for a thorough and helpful review of our manuscript, and would like to say that we really appreciate the time the reviewer invested for helping us improving the manuscript.

We conducted the study off season because exactly because it rarely has been done and we were hoping to come with this to an understanding of $N_2$ fixation over the course of the year. Most studies are conducted in and around the bloom seasons, but fall/ winter studies are rarely presented. We, however, find it important to do so, and hope this study can contribute to a better assessment of $N_2$ fixation over the course of the year, and possibly inform models by doing so. In order to clarify this point, we provided an explanation in L.89 of the revised version of the manuscript:

'To now explore the impact and importance of salinity for diazotrophy in the Baltic Sea and in order to complement existing datasets on N2 fixation from high-productivity seasons, we carried out a survey…'

I think the combination of genes and transcripts is interesting, but I wonder if the transcripts can be somehow quantitative. Maybe you can present also transcripts per gene copies so its normalized to abundance? I think this particular data is what is novel with this study and should be lifted in the manuscript and aims. Especially UCYN-A has not been widely studied in the Baltic Sea and not this late in the season.

AR: We used a qPCR (quantitative real time PCR) for both gene copies and transcript copies, which is a fully quantitative method measured against a standard dilution series with known copy numbers. To present transcripts per gene copies is a good idea and a valid point, we now added a figure showing the transcriptional activity in that way.

I am a little bit confused by the use of the bubble method here for N2 fixation as I thought this was not used any more due to the risk of underestimation (e.g., White 2012) and recent studies in the Baltic Sea has used dilution method (e.g., Klawonn et al. 2016). I think this should be discussed in the manuscript and not lifted as an advantage as it is now. Can it be that rates were underestimated?

AR: True, and some of the authors were directly involved in developing the pre-dissolution method, which has later been improved by Klawonn et al. The initial purpose of the study was indeed to compare our data, however, to older datasets to reconstruct annual cycles in a Baltic Sea regional ocean model, therefore, we aimed at keeping the data comparable to older studies. Rates, especially those at the lower end of $N_2$ fixation according to Grosskopf et al might be underestimated, therefore, the estimate here is conservative and not fully quantitative. On the other hand, Klawonn et al, 2015 report a potential impact of trace metal contamination leading to an overestimation when using the

dissolution method, our aim was to then rather remain conservative and not risk overestimation. We added the following explanation to L. 155 ff of the manuscript:

'We used the bubble addition method (Montoya et al., 1996) to ensure comparability to previous studies from the Baltic Sea, although we are aware that more recent studies (Klawonn et al., 2016) used the dilution method. The latter study used the dissolution method, which is prone to trace metal contamination and might lead to an overestimation of $N_2$ fixation (Klawonn et al., 2015; Benavides et al, 2016), which we wanted to avoid. Our rates, however, seem to be within the range of all earlier studies, despite they represent a conservative assessment and possibly an underestimation (White et al., 2012, Grosskopf et al, 2014)'

References:

Benavides, M., et al. (2016), Basin-wide N2 fixation in the deep waters of the Mediterranean Sea, Global Biogeochem. Cycles, 30, 952– 961, doi:10.1002/2015GB005326.

Klawonn, I., G. LaviK, and P. Böning (2015), Simple approach for the preparation of 15N2-enriched water for nitrogen fixation assessments: Evaluation, application and recommendations, Front. Microbiol., doi:10.3389/fmicb.2015.00769.

The statistics needs some work in the paper, better describe the PCA in the text and why you have two graphs presenting what seems like contradicting results. As it is now it is difficult to tell apart what is correlated to what since all arrows has the same colors and point in different directions in A and B. Maybe it is better to run an NMDS analysis for stations and taxonomic groups correlated with environmental factors (as arrows on top), and only one panel? With significant arrows in one color and non-significant in another.

AR: The PCA is now described in more detail. The colors don't have any information; however, the directions and lengths of the arrows do. The directions tell us which components are positively (same directions) or negatively (opposite directions) correlated. The length tells us the "contribution" to the explained variance. Thus, the longer the arrows are the more weight the explanation of the variance we have. Lastly, we show the panels A and B of the figure because a PCA plot is a three-dimensional analysis with x number of components. In A, we show PCA1 and 2. In B we show another perspective using PCA3. The text has been modified to make it better understandable.

I suggest that the manuscript need major revision before being considered for publication.

Detailed comments

Lines 7-8, change to "decreased salinity due to altered precipitation related to climate change"?

AR: Done.

Line 8, clarify N2 by including nitrogen (N2) fixing?

AR: Done.

Line 17, Nodularia spumigena?

AR: Yes, corrected.

Lines 18 and 19, gene copies? Cells? Filaments?

AR: Yes, this should be gene copies, which has been revised.

Line 20, are you sure they are $N_2$ fixers or should you say potential before?

AR: The $N_2$ fixers were detected using specific key functional marker genes for either nifH or anf/vnfD which were amplified using PCR. Based on this in combination with a bioinformatic (including individual BLAST search on NCBI and alignment with related nitrogenase sequences) and phylogenetic analysis they were identified as $N_2$ fixers. Therefore, genetically those microbes are $N_2$ fixers, and this is typically used as a basis to call them $N_2$ fixers. If we now of course think about $N_2$ fixers being only organisms which are actively fixing $N_2$ we would have to add 'potential'. It is therefore rather a linguistic but interesting problem. We would therefore be tempted to leave it like it is.

Line 20, I think statistical testing here is redundant. Either say what test or just skip it and start the sentence from "salinity was identified".

AR: Done.

Line 22, similarly significant? Either it is significant or not?

AR: Done.

Line 38, change to severely affects?

AR: Done.

Line 44, is this how the reference should look like?

AR: No, we reformatted the reference.

Line 46, remove the comma after both?

AR: Done.

Line 66, Anabaena/Dolichospermum if often referred to as spp. since its not only one species. You should state Klawonn et al. 2016 or similar study here when saying these three carry out the N2 fixation.

AR: Yes, this has been adjusted, Klawonn et al has been added, here.

Line 68, maybe also mention rates from newer studies such as Klawonn et al. 2016.

AR: Done.

Line 69, I don't understand this huge deviation between what the fix and consume, the 35% in Ploug et al. would not explain that big difference in rates.

AR: Thanks for this comment, indeed there seems to have been a calculation error which has been corrected, now. The rates are now 115-295 nmol N $L^{-1}$ $d^{-1}$ fixed and 33.33 – 85.74 nmol N $L^{-1}$ $d^{-1}$ leaked. Roughly 30% leaked out which is more in line with Klawonn et al. 2016, a dataset that is now also presented along with our data in the text. Further, a deviation could result from different assessments used. We were referring to Larsson et al. (2001), where $N_2$ fixation rates were estimated based on the increase in total nitrogen in the upper 20 m of the water column, the incorporation of N into biomass is, however, based on C:N ratio. Thus, the estimates presented by Ploug et al might indeed be more accurate. We adjusted the respective text, it now reads:

'… contributed massively to $N_2$ fixation with rates of up to 115 – 295 nmol N $L^{-1}$ $d^{-1}$ (Larsson et al., 2001). At the same time, cyanobacteria consume only a fraction of the N they fix, with estimates in the range of 33.33 – 85.72 nmol N $L^{-1}$ $d^{-1}$ leaking out (Janson et al., 1994; Larsson et al., 2001; Wasmund, 1997) roughly in line with estimates given in Klawonn et al. (2016) based on isotope measurements and cell-specific $N_2$ fixation experiments of *Aphanizomenon sp* and *N. spumigena*, which demonstrated a release of a substantial fraction of up to 35% of fixed N available for other primary producers (Ploug et al., 2010, 2011).'

Line 72, missing period.

AR: Added.

Line 81, what unit?

AR: This refers to salinity, which is unitless now, and also was given without a unit in the paper referred to.

Ln 82, how does it affect the nitrogenase activity?

AR: We can only guess that there might be a physiological optimum for the nitrogenase depending on the salinity in the medium or the environment. It might also have to do with the impact of salinity on gas solubility. However, there is no clear explanation provided.

Line 86, abbreviate to N. spumigena after first mentioned. I think you should broaden the range, Aphanizomenon is extremely common in the Baltic Proper with salinities around 5-6, with higher densities than the Bothnian Sea.

AR: Done. The information on the salinity range has been revised.

Line 87, change "speak for" to indicate?

AR: Changed.

Line 89, include Southern Baltic Sea?

AR: Done.

Line 90, why did you choose a low productive season?

AR: Please see the above comment, we added a text to explain this to the manuscript.

Line 91, what do you mean by controls?

AR: We were referring to any environmental parameter, or factor, which could have a potential impact on $N_2$ fixation. This has now been clarified in the text:

'…to explore how various environmental parameters impact on Baltic Sea $N_2$ fixation, and…'

Lines 89-92, I think you should revise this aims section as you need to argue for why you choose the low productive season and also that you actually look at genes and transcripts which is really cool. That salinity affects the diazotrophic community composition is already known, but it is novel that you here look at other diazotrophic taxa than the heterocystous ones, I think this should somehow be told here too.

AR: Thanks for this encouraging comment, we emphasized that more now, the section now reads:

'To now explore the impact and importance of salinity for diazotrophy in the Baltic Sea and in order to complement existing datasets on $N_2$ fixation from high-productivity seasons, we carried out a survey including chemical profiling, and rate measurements in the low productive fall season. The focus of this study is a molecular genetic assessment of the diazotrophic community diversity and genetic activity including heterocyst and unicellular cyanobacteria, heterotroph diazotrophs and diazotrophs carrying alternative nitrogenase genes.'.

Line 97, change on to using and remove "to the Baltic Sea"?

AR: Done.

Line 102, both ammonium, nitrate, and nitrite?

AR: This has been clarified, it was only nitrate and nitrite.

Line 110, change to fixed?

AR: Done.

Lines 112-113, revise so that it is clear that the 0.5-1 L was filtered onto the membrane filters and change filter to plural. On all stations and depths?

AR: Done. We added a table with all stations and depths, and indicated, which samples were taken, and where.

Line 115, it is not clear to me how these samples were collected, all from the 0.5-1 L filtered samples?

AR: Yes, this refers to the sentence before. We changed the sentence to clarify this to:

'For nucleic acid extraction, the 0.22 µm pore sized Millipore filters (see above) were flash-frozen…'

Line 123, one from each station? Please clarify how you came up with 15. Which depth?

AR: We hope, this will now be clear from the new table added, where we indicated, which samples we referred to.

Lines 132-135, why these random depths?

AR: We chose the depths based on the water column biogeochemistry, this included the chlorophyll max, the halocline etc. We added a table showing the samples taken and the additional parameters.

Line 149, the statistics section is usually found at the end of the method section?

AR: We moved it to the end of the methods part.

Line 150, I think you need to provide more details on the PCA here, what are the different components in the figure? What is the difference between figure A and B?

AR: See also our comment above, we added the following explanation:

'To maximize the amount of explained variance of the dataset, the figure included both component one versus two (Fig. 9A), and one versus three (Fig. 9B).'

Line 155, why these stations and which depths?

AR: The stations were chosen so to cover all different regions covered by the cruise track to obtain information on $N_2$ fixation throughout the various regimes present there. Because incubations are run over 24 hours, some stations couldn't be covered for practical reasons including a limitation of time and manpower. The exact depths are now also included in the newly added sampling table.

Lines 156-157, I do not fully agree with this since lately this method has not been used due to underestimation risks (e.g., Klawonn et al. 2016).

AR: Please see our response above, we generally agree that there are better methods available, and we should indeed next time use the bubble addition method and the pre-dissolution method in parallel. However, this was not possible during this particular survey for practical reasons, mostly because we had only one person on board.

Line 157, top-filled?

Yes, the protocol was the same as in Löscher et al., 2014, an information that has been added, now.

Line 160, did you measure this or is it based on calculations?

AR: Calculated, the information has been added.

Line 162, were they incubated for the same time? Any labelled T0 samples collected to ensure free of contamination?

AR: We didn't collect labelled T0 samples, and this is also never part of our typical protocol. However, we collected unlabeled T0 samples.

Line 168, feels like a few words are missing, the typical Baltic Sea gradient I guess?

AR: Yes, we changed it to '…the typical salinity gradient in the Baltic Sea…'

Line 170-171, this is very low temperatures for Nodularia.

AR: Correct, still they are around.

Line 178, lower than what? Expected?

AR: This is a comparison to the south-western part. We clarified this in the text.

Line 180, you refer to DIN in the methods? Please be consistent.

AR: We corrected this throughout the manuscript to keep it consistent.

Lines 184-185, higher than the report or as compared to other locations?

AR: Higher than the report, this has been changed to 'In comparison to those reported concentrations, some of our observed nutrient concentrations in the Bornholm and Eastern Gotland Basins….'

Line 195, or temperature?

AR: Changed.

Line 196, that does more sound like they have more N than C not N depletion?

AR: Exactly, that's why we talk about N repletion.

Line 199, you have another order in the methods, with N2 fixation and community, consider keeping it consistent?

AR: True, we moved the rate measurements in the methods section to an earlier place.

Line 203, what is your detection limit?

AR: $N_2$ fixation rates were calculated with the equations of Montoya et al. (1996). Considering the particulate nitrogen linearity limit of the mass spectrometer (2. 2 µg N), 3 times the standard deviation of our background (time zero atom % 15N) values, our usual filtration volume, and incubation time, our volumetric $N_2$ fixation rate detection limit would be 0.03 nmol N $L^{-1}$ $d^{-1}$.

Line 204, did this occur together with high Chl a concentration or cyanobacteria abundance?

AR: No, not clearly, but indeed it might be that we caught one 'loaded' filament on our filter. This is, however, only a speculation.

Line 208, lower the "2".

AR: Done.

Line 208, maybe expand here connect with lines 212-218? Maybe move the isotope data to the niche discussion?

AR: Done.

Line 209, stick to past tense, "was supported".

AR: Done.

Line 216, abbreviate to N. spumigena.

AR: Done.

Line 217, low light when? As compared to what?

AR: We changed this to '…lower light intensities during fall and winter…'

Line 219, if you state low rates, maybe give examples of high rates from the area or put in some kind of perspective.

AR: We rephrased this to 'low C fixation rates compared to rates in the micromolar range as typically detected in the Baltic Sea (e.g., Klawonn et al., 2016) …'

References:

Klawonn, I., Nahar, N., Walve, J., Andersson, B., Olofsson, M., Svedén, J.B., Littmann, S., Whitehouse, M.J., Kuypers, M.M.M. and Ploug, H. (2016), Cell-specific nitrogen- and carbon-fixation of cyanobacteria in a temperate marine system (Baltic Sea). Environ Microbiol, 18: 4596-4609. https://doi.org/10.1111/1462-2920.13557

Line 222, maybe mention where the rest of the N might come from?

AR: We cannot make a clear statement, here, but assume it would be either recycling, land runoff, riverine or atmospheric sources.

Line 236, it is very common in the Baltic Proper, starting early in the season, with salinity of 5-6 (e.g., Svedén et al. 2015 in FEMS) and dominating N2 fixation (Klawonn et al. 2016). Saying that it prefers as low as 0-2 is not really true here.

AR: Changed.

Line 238, I think you rather mean Dolichospermum? Although it used to be called Anabaena it has been referred to as Dolichospermum since 2009 (Wacklin) and then you can also refer to Olofsson et al. 2020 and Klawonn et al. 2016 here. You refer to it in line 66.

AR: Changed.

Lines 423-427, how does gene copies relates to cell numbers?

AR: We are not sure if the line number is correct here (maybe 243-247?). However, the assumption is that one gene copy is present in one cell. There are some known deviations from this, but we are not aware of such deviations being described for the clades we identified.

Line 261, check the reference layout.

AR: Changed.

Line 276, how can genes and transcript be related to cell numbers?

AR: Gene copy numbers are related to cell numbers; transcript abundances are not.

Line 298, do you mean Olofsson et al. 2020 here? Karlberg and Wulff was a laboratory study. Or if you refer to several studies in the sentence then include them, now it sounds like Karlberg and Wulff has done everything in the sentence including modeling.

AR: We added Olofsson et al.

Line 313, maybe specify where this freshening would be beneficial to Nodularia? Since freshening in the northern Baltic sea will have an opposite effect since its already very low.

AR: We changed this sentence (see also comments of reviewer 2) to

'In case of a potential future freshening of the upper water column (Liblik and Lips, 2019) in combination with increased $PO_4^{3-}$ availability both through land-derived influx and through phosphorous mobilization via bottom-water anoxia (Ingall and Jahnke, 1997; Vahetra et al., 2007; Gustafsson et al, 2017; Stigebrandt and Anderson, 2020), $N_2$ fixation by e.g. Nodularia will likely increase in the region covered by our cruise.'

Line 324, maybe include south here?

AR: Done.

Line 725, Olofsson et al. 2021 is missing from the reference list?

AR: We included the reference now.

Figure 1, maybe have the three figs in a row in the same size instead? Why is it called last? Can you mark those stations where N2 fixation were measured? And explain in the legend.

AR: We re-arranged the figure. With 'last' we meant the deepest samples measured at each station. It is now clarified in the figure legend.

Figure 2, maybe include 20°E as well since this on the map in Fig. 1? Number of samples?

AR: Done.

Figure 3, "pres [db]" need to be better described. This figure feels a bit redundant, maybe move to supplementary? Number of samples?

AR: The description has been changed to 'Distribution of chlorophyll (Chl [µM] along the natural pressure gradient (press [db]) over the year, data derived from…''. The figure has been moved to the supplement. The number of samples was 5 depths x 12 months = 60 samples per basin, this information has been added.

Figure 4, the text on the axes is far too small. Also, the information within the graph-window. A lot of the discussion part of the text in the legend should be in the manuscript instead. Why is it limited to those stations? Number of samples?

AR: We increased the size of the text on the axes, and of the information text. The information on numbers of samples has been added. The aim was to follow the salinity gradient. However, as it was a short cruise (11 days, with 1.5 day docking at Gotland) we needed to reduce the number of stations.

Figure 5, consider stretch the x-axis so that the numbers is visible, now it almost overlaps. I would suggest that you write out carbon fixation in the legend, now it looks like it say Figure 5 C. This legend also consists of much results that should go into the manuscript instead. Why limited to those stations? Number of samples?

AR: Done. We left the result description in the legend, it might be a bit repetitive, but we believe it can be helpful to read the figure without jumping back to the text. The number of samples has now been added. Same explanation as above, regarding the limited amount of station.

Figure 6, the text is too small to be able to read, can you make it larger? Number of samples? Move to supplementary since you already have so many figures?

AR: The text size has been increased; the number of samples has been added to the legend. The figure is, however, a key figure to the manuscript, we would therefore like to keep it in the main text.  The missing information has been added to the legend.

Figure 7, move to supplementary since you already have so many figures?

AR: We would like to keep this figure in the main text as it is a key figure. Figure 3 has been moved to the supplement.

Figure 8, consider using colors instead, now its hard to distinguish. Can you maybe make also a graph with transcripts per gene copies so its normalized to abundance? Number of samples?

AR: Done.

Figure 9, what is components one, two, and three? I don't see the differences between the figures and what has been done? Also how can for example N2 fixation and Nodularia point in different directions on B while being in the same directions in A? What do you mean by the center of each station?

AR: See also our comment above. $N_2$ fixation and Nodularia point in different directions based on two different angles. Yes, it might be contradictory which suggests that Nodularia is not solely responsible for $N_2$ fixation. "center of each station" is the larger dots which makes up a center for the three datapoint from each station.

Table 1, how were these numbers extracted? For example, 0.36 and 0.37 mmol N m-2 d-1 in the table is much lower than reported in Olofsson et al. 2021 figure 5? Provide ranges when you have, Klawonn et al. 2016 for example also have many measurements so you should be able to provide a range or mean value with SD. Why does some have ranges and some means and some just one value? Also use the same number of significant digits where you can, Rinne et al. for example has an absurd number of digits?

AR: The table has been revised according to your suggestions.

Also, numbers before 2001 were extracted from Wasmund et al. 2001 and calculated into mmol N L$^{-1}$ d$^{-1}$. Numbers from 2001 to present were collected by literature search and converted into mmol N L$^{-1}$ d$^{-1}$ for comparability. The numbers from Olofsson et al. 2021 were calculated from e.g., $384 \pm 74$ kt yr$^{-1}$ to mmol N L$^{-1}$ as follows (a dimensional error was corrected, too):

$$\frac{374 \text{ kt year}^{-1}}{(210000 \ km^2 * 1000000 \ m^2 \ km^{-2})} = 1.82 * 10^{-9} \ kt \ m^{-2} \ year^{-1}$$

$$\frac{1.82 * 10^{-9} \ kt \ m^{-2} \ year^{-1}}{365 \ days \ year^{-1}} = 5 * 10^{-12} \ kt \ m^{-2} \ d^{-1}$$

$$5 * 10^{-12} \ kt \ m^{-2} \ d^{-1} * 10^9 \frac{g}{kt} = 0.05 \ g \ m^{-2} \ d^{-1}$$

$$\frac{0.05 \ g \ m^{-2} \ d^{-1}}{14 \ g \ mol^{-1}} = 0.0035 \ mol \ m^{-2} \ d^{-1}$$

$$0.0035 \ mol \ m^{-2} \ d^{-1} * 1000 \ mmol \ mol^{-1} = \mathbf{3.5 \ mmol \ m^{-2} \ d^{-1}}$$

Wasmund et al 2001, Evidence of nitrogen fixation by non-heterocystous cyanobacteria in the Baltic Sea and re-calculation of a budget of nitrogen fixation

Table 2, is these mean values for those studies? Provide more details in the legend. There should be more studies providing this for the Baltic Sea? Are they from the same season? Species?

AR: Surely there are other studies from the Baltic Sea, which are collected in Table 1. The purpose of this table was to set our data into a more global context for which we used studies having provided this exact analysis. We added the information on the season to the table. We also changed the figure legend to 'Contribution of $N_2$ fixation to primary production (PP) in different ocean basins, values represent mean values for the respective regions.'

Additional changes: We added a personal acknowledgment to Dr Kuosa and an anonymous reviewer.

---

## Author Comment (AC2)

This is a good paper with many interesting observations. Though the text is fine, I would like to focus it more - like the title suggests - to salinity/N-fixer communities. The chapter on Nitrogen fixation rates (3.2.) does not represent typical situation due to the timing of the samplings. nitrogen fixation rates are generally low as the authors have also shown in the text. The chapter could be condensed, and I do not find novelty in the summarizing Table 1. However, the other parts of the article are fine. Introduction should tell us more about the UCYN-A organisms and their ecology as this is one of the major findings in the article.

Authors' response (AR): We are glad to read that reviewer 2 finds our study interesting, we are grateful for the time Dr Kuosa invested into improving our study, for the comments and suggestions. In the revised version, we aimed at setting a stronger focus on salinity/N-fixer communities. We further added a more detailed introduction to the ecology of UCYN-A to l. 68:

'Additionally, the small, unicellular cyanobacterial symbiont UCYN-A has been detected in the Baltic Sea (Bentzon-Tilia et al., 2015). This cosmopolitan diazotroph has previously been shown to be abundant throughout most marine systems (Zehr et al., 2016; Tang et al., 2019), and to substantially contribute to N2 fixation rates (Martinez-Perez et al., 2016; Mills et al., 2020).'

References:

Martínez-Pérez C, Mohr W, Löscher CR, Dekaezemacker J, Littmann S, Yilmaz P, Lehnen N, Fuchs BM, Lavik G, Schmitz RA, LaRoche J, Kuypers MM. The small unicellular diazotrophic symbiont, UCYN-A, is a key player in the marine nitrogen cycle. Nat Microbiol. 2016 Sep 12;1(11):16163. doi: 10.1038/nmicrobiol.2016.163.

Mills MM, Turk-Kubo KA, van Dijken GL, Henke BA, Harding K, Wilson ST, Arrigo KR, Zehr JP. Unusual marine cyanobacteria/haptophyte symbiosis relies on N2 fixation even in N-rich environments. ISME J. 2020 Oct;14(10):2395-2406. doi: 10.1038/s41396-020-0691-6. 2020

Tang, W., Wang, S., Fonseca-Batista, D. et al. Revisiting the distribution of oceanic N2 fixation and estimating diazotrophic contribution to marine production. Nat Commun 10, 831 (2019). https://doi.org/10.1038/s41467-019-08640-0

Zehr JP, Shilova IN, Farnelid HM, Muñoz-Marín MD, Turk-Kubo KA. Unusual marine unicellular symbiosis with the nitrogen-fixing cyanobacterium UCYN-A. Nat Microbiol. 2016

The table summarizing previous studies might not be novel, but we believe that it is helpful for seeing our data in context of other seasons and studies. We however, moved it to the supplementary material.

I have some detailed comments:

Line 30: The Baltic Sea covers an area of 415000 km2 with a permanent halocline preventing vertical mixing, oxygen (O2)-depleted waters in the deeper basins and coastal systems, accompanied with the occasional accumulation of hydrogen sulfide (H2S) and ammonium (NH4+) below the chemocline.

Northern deep basins (Åland Sea and Bothnian Sea) do not have a permanent halocline.

Phosphate accumulation should be mentioned.

AR: We clarified those points and included the phosphate accumulation; the text in l. 30 ff now reads:

'The Baltic Sea covers an area of 415000 km2 with a permanent halocline in the Baltic Sea proper, preventing vertical mixing, oxygen (O2)-depleted waters in the deeper basins and coastal systems, accompanied with the occasional accumulation of hydrogen sulfide (H2S) and ammonium (NH4+) below the chemocline (…). It is further challenged by a high land-derived influx of phosphorous leading to a substantial internal surface water and sedimentary phosphorous load (Gustafsson et al, 2017; Stigebrandt and Anderson, 2020).'

References:

Gustafsson, E., Savchuk, O.P., Gustafsson, B.G. et al. Key processes in the coupled carbon, nitrogen, and phosphorus cycling of the Baltic Sea. Biogeochemistry 134, 301–317 (2017). https://doi.org/10.1007/s10533-017-0361-6

Stigebrandt, A., Andersson, A., The Eutrophication of the Baltic Sea has been Boosted and Perpetuated by a Major Internal Phosphorus Source, Frontiers in Marine Science, vol. 7, 2020, p. 996 , https://www.frontiersin.org/article/10.3389/fmars.2020.572994

Line 44: Reference 'Capone, Douglas G; Carpenter, 1982;' is atypically written compared to others. Capone et al. 1982?

AR: We formatted this reference.

Line 66: 'heterocytous' originating from a cell (cyte) is preferred instead of 'heterocystous' (cyst).

AR: Changed.

Line 66 and 70: '*Aphanizomen*' should be '*Aphanizomenon*'.

AR: Changed.

Line 73: 'available for primary production' = 'available for other primary producers'?

AR: Changed.

Line 86: '(e.g. the Bothnian Sea) with a salinity of 0-2' There may be an error here as the Bothnian Sea is closer to 5 in its salinity.

AR: Changed.

Line 87 onwards: see Laamanen, M. J., Forsstrom, L., & Sivonen, K. (2002). Diversity of *Aphanizomenon flos-aquae* (cyanobacterium) populations along a Baltic Sea salinity gradient. Applied and Environmental Microbiology, 68, 5296-5303. https://doi.org/10.1128/AEM.68.11.5296-5303.2002

AR: This is very helpful- it has been included throughout the manuscript.

Line 105: DIN analysis includes both ammonium and nitrate (+nitrite)?

AR: It only included nitrate and nitrite, we clarified this.

Line 164: 'dried' can mean many different methods with different end results. What is used here?

AR: We dried them at 65 degrees C overnight, this information has now been added.

Line 177: 'basin' with capital 'b'.

AR: Changed.

Line 182: The results are given as NOx instead of DIN in the methods?

AR: This was also a comment from the other reviewer, it has been clarified what we mean (see above) and made consistent throughout the text.

Line 184: 'The detected somewhat higher nutrient concentrations in the Bornholm and Eastern Gotland Basins could result from a decaying phytoplankton bloom, decreased microbial activity or increased eutrophication.' This should lead to elevated ammonium concentrations.

AR: Correct, we unfortunately don't have ammonia measurements, but we included a statement on this into the text:

'The detected somewhat higher nutrient concentrations in the Bornholm and Eastern Gotland Basins could result from a decaying phytoplankton bloom releasing nutrients including ammonia, decreased microbial activity or increased eutrophication.'

Line 192: Only NOx is discussed. Did the samples have notable concentrations of ammonium?

AR: See above, we unfortunately do not have ammonia data from this cruise.

Line 303: 'Moreover, a very recent study showed that ocean acidification has an impact on the diazotroph community composition and can decrease N2 fixation rates in the subtropical Atlantic Ocean (Singh et al., 2021).' These N-fixers (*Trichodesmium*) are very different in their ecology. pH may have an effect on their growth, which is then reflected by their N-fixation capacity, not that pH directly affects N-fixation.

AR: Yes, this is a sensitive thought and difference, we clarified it in the text, which now reads 'Moreover, a very recent study showed that ocean acidification has an impact on the diazotroph community composition and ecology and can decrease bulk $N_2$ fixation rates in the subtropical Atlantic Ocean (Singh et al., 2021).'

Line 313: 'In case of a future freshening of the upper water column…' This conclusion should be tied with basin-wide P-dynamics as it also affects the future of cyanobacterial blooms. I would propose using 'potential' in this paragraph.

AR: Good point, we changed this sentence to 'In case of a potential future freshening of the upper water column (Liblik and Lips, 2019) in combination with increased $PO_4^{3-}$ availability both through land-derived influx and through phosphorous mobilization via bottom-water anoxia (Ingall and Jahnke, 1997; Vahetra et al., 2007; Gustafsson et al, 2017; Stigebrandt and Anderson, 2020), …'.

Additional reference:

Vahtera, E., Conley, D. J., Gustafsson, B. G., Kuosa, H., Pitkänen, H., Savchuk, O. P., et al. (2007). Internal ecosystem feedbacks enhance nitrogen-fixing cyanobacteria blooms and complicate management in the Baltic Sea. *Ambio* 36, 186–194. doi: 10.1579/0044-7447(2007)36[186:IEFENC]2.0.CO;2

Additional changes: We added a personal acknowledgment to Dr Kuosa and an anonymous reviewer.

---

## Author Response (AR2)

**Reviewer 1**

I think the authors have made a good job revising the manuscript and I only have a few minor comment on the current version, but other than that I suggest it can be accepted for publication.

Authors' response (AR): We thank the reviewer for taking the time to go through the manuscript and time invested to improving the manuscript.

Maybe add southern to the title?

AR: Done

Lines 9, 25, 355 and more, I have not seen heterocytous before and would prefer heterocystous, but I see that reviewer 2 prefers the first one so that is up to you. I would argue they are generally called heterocysts.

AR: Heterocytous has been change to heterocyst

Lines 38-39 and 351, phosphorous should read phosphorus.

AR: Done

Lines 78-79, change measure to measured? Add a dot after spp.

AR: Done

Line 95, I still think you need to increase the range of Aphanizomenon salinities to either <10 or optimum at around 5 (if looking at Lehtimäki et al. 1997 that you refer to), it grows a lot on the northern Baltic Proper with salinities around 5-6 and is overall dominating in mainly salinities between 5-8 (Olofsson et al. 2020). Also you refer to a higher range at line 268.

AR: The sentence has been changed to the following:

"Typically, Aphanizomenon sp. dominates less saline waters (e.g., Bothnian Sea) and grow best with salinities around 5 (Lehtimäki et al., 1997). Aphanizomen sp., however, has also been frequently reported to grow at salinities reaching up to 5-8 such as the northern Baltic Proper (Olofsson et al., 2020). N. spumigena prefers the higher saline waters in southern part (e.g. the Southern Baltic Proper) with an optimum growth in salinities of 8-10 (Lehtimäki et al., 1997; Rakko and Seppälä, 2014)."

Lehtimäki, J., Moisander, P., Sivonen, K. and Kononen, K.: Growth, nitrogen fixation, and Nodularin production by two Baltic Sea cyanobacteria, Appl. Environ. Microbiol., 63(5), 1647–1656, doi:10.1128/aem.63.5.1647-1656.1997, 1997.

Olofsson, M., Suikkanen, S., Kobos, J., Wasmund, N. and Karlson, B.: Basin-specific changes in filamentous cyanobacteria community composition across four decades in the Baltic Sea, Harmful Algae, 91(October 2019), 101685, doi:10.1016/j.hal.2019.101685, 2020.

Rakko, A. and Seppälä, J.: Effect of salinity on the growth rate and nutrient stoichiometry of two

Baltic Sea filamentous cyanobacterial species, Est. J. Ecol., 63(2), 55–70, doi:10.3176/eco.2014.2.01, 2014.

Line 141, what final percentage of 13C?

AR: Final concentration was around 3.8 atom %. This has been added to the text. It now says: "...and 10  $\mu$ g mL-1 H13CO3 (approximately 3.8 atom %)."

Lines 219 and 224, mol:mol? Maybe clarify by including in brackets. AR: It is  $[\mu mol L^{-1}:\mu mol L^{-1}]$  and has been added to the text.

Line 315, what do you mean by "at the sampling locations monitored during our cruise"?

AR: Normally, *Rhodopseudomonas* require anoxia to fix nitrogen. At the sample location monitored, we had oxic condition. So, *Rhodopseudomonas* might not utilize the alternative nitrogenases. Nevertheless, I see that the sentence is confusing. I decided to delete the sentence, since I do not think it offers any significant relevance in the context.

Line 331, remove modelling? Neither Karlberg (laboratory study) or Olofsson (monitoring data) include modelling, but if you want a study of cyanobacteria modelling in the Baltic Proper you might want to refer to Hieronymus et al. 2020 in Biogeosciences but I am not sure it's a relevant reference here.

AR: Remove "and modelling approaches". It now reads: "While a study based on a compiled dataset (1979-2017) indicated that salinity does not affect the biovolumes of the filamentous N. spumigena but rather species-interactions (Karlberg and Wulff, 2013; Olofsson et al., 2020)"

Karlberg, M. and Wulff, A.: Impact of temperature and species interaction on filamentous cyanobacteria may be more important than salinity and increased pCO2 levels, Mar. Biol., 160(8), 2063–2072, doi:10.1007/s00227-012-2078-3, 2013.

Olofsson, M., Suikkanen, S., Kobos, J., Wasmund, N. and Karlson, B.: Basin-specific changes in filamentous cyanobacteria community composition across four decades in the Baltic Sea, Harmful Algae, 91(October 2019), 101685, doi:10.1016/j.hal.2019.101685, 2020.

Line 386, feel free to include Malin Olofsson (or Dr. Olofsson to be consistent) here as well instead of anonymous, it will anyhow be listed once the paper is accepted.

AR: Done. The sentence now says: "We thank Dr. Kuosa and Dr. Olofsson for reviewing the paper."

Fig. S1. Is this really relative abundance? It looks more like some type of count measurement, and if it is it should have a unit. If relative I would assume it should be presented as % or 0-1? Based on replicates or one sample each? If replicates please include number of n.

AR: The figure has been remade so it shows relative abundance in %.

The references in table S3 should be listed somewhere? For example I still don't see Olofsson et al. 2021 in the reference list? Also Wasmund et al. 2001 and 2005?

AR: They have been added now. References were accidently deleted when moving the table to supplementary.

Additional changes.

AR: Line 132 0.05 % 15N2 was changes to 0.8%. Saw a error when calculating the final % of 13C.

**Editor**

Dear authors

Please revise the manuscript according to the comments by reviewer #1. In addition, the text parts about future projections of salinity in the Baltic Sea should be updated. According to recent research, projected global sea level rise might counteract the freshening due to projected increases in precipitation (see the review article https://esd.copernicus.org/articles/13/159/2022/). Furthermore, projections are rather uncertain and it is unknown whether future salinity will increase or decrease (Meier et al., 2021).

Best wishes Markus Meier

Reference:

Meier, H. E. M., C. Dieterich, and M. Gröger, 2021: Natural variability is a large source of uncertainty in future projections of hypoxia in the Baltic Sea. Commun. Earth Environ. 2, 50, https://doi.org/10.1038/s43247-021-00115-9

AR: The following text has been added to the text part about the future projection of salinity in the Baltic Sea.

"Latest studies, however, shows that projection of increased sea level rise counteracts freshening events in the Baltic Sea making it uncertain whether salinity increases or decreases in the future (Meier et al., 2021, 2022). Regardless, potential changes in salinity might impact the diazotrophic community as described above. The uncertainty in changing salinity makes it unclear in which extent diazotrophs might be impacted."

Meier, H. E. M., Dieterich, C. and Gröger, M.: Natural variability is a large source of uncertainty in future projections of hypoxia in the Baltic Sea, Commun. Earth Environ., 2(1), doi:10.1038/s43247-021-00115-9, 2021.

Meier, H. E. M., Dieterich, C., Gröger, M., Dutheil, C., Börgel, F., Safonova, K., Christensen, O. B. and Kjellström, E.: Oceanographic regional climate projections for the Baltic Sea until 2100, Earth Syst. Dyn., 13(1), 159–199, doi:10.5194/esd-13-159-2022, 2022.